# The effect of fertility on female labour force participation in Tanzania

**Aidan Laurent Sunday, Robert Michael Lihawa, Eliaza Mkuna** *

Department of Economics, Faculty of Social Sciences, Mzumbe University, Morogoro Tanzania

* eliazamkuna@hotmail.com

## Abstract

Country's economic growth depends among other factors on the extent to which labour particularly female labour force participates on economic growth enhancing activities. Being the largest contributor in economic activities particularly agriculture in developing countries (over 50%), their participation enables economies to grow in response to higher labour inputs injected. As an outcome, as countries develop; women's capabilities typically improve as well, whereas social constraints weaken, which enables females to participate on work outside the home. However, the existing literature on this topic is scant and has mixed results. This study sought to analyse female fertility rate and its influence on provision of labour in Tanzania using females aged 15–49 years from the Tanzania Demographic and Health Survey 2015–2016. The study used instrumental variable-probit and a two-stage residual inclusion as methods of analysis. Results showed that, an increase in female's fertility rate reduces participation of females in provision of market labour by about 1.1–13%. Similarly, household size, education, contraceptive use, self-employment of their husbands and residing in rural areas was associated with increased participation while female's age exhibited an inverted U-shaped relationship with female participation. The results imply that, to foster a more sustainable female participation in labour force, family planning, educating females as well as fostering self-employment and improving rural infrastructures is inevitable.

## 1. Introduction

One of the major hindrances in ensuring worldwide equitable and inclusive labour market is the gap existing between males and females when it comes to accessing labour market opportunities. According to ILO, these gaps emanating in the inequality in the market labour force involvement among males and females costs about 15% of the global gross domestic product (GDP) [1]. While there are many potential reasons behind these gender gaps; labour market existing laws and regulations, fertility as well as unequal access to education definitely plays a huge role. According to the World Bank, existing laws restricts over 2.7 billion females from accessing the same labour market opportunities as that of men [2]. Legal frameworks in 104 countries in the world prohibits women from participating in specific works, in 59 economies these legal frameworks do not protect women from sexual harassments at work and in 18

**Data Availability Statement:** This study used third party data from the 2015-2016 Tanzania Demographic and Health Surveys (DHS) publicly available through The DHS Program website (https://www.dhsprogram.com/data/available-

datasets.cfm). The authors confirm that others would be able to access these data in the same manner as themselves. The authors also confirm that they did not have any special access privileges.

**Funding:** The author(s) received no specific funding for this work.

**Competing interests:** The authors have declared that no competing interests exist.

countries husbands are legally entitled to stop their wives from working [2]. Regarding fertility, labour force involvement among females in the Caribbean and Latin American countries rose from around 45% in 1995 to about 53% in 2015 primarily due to significantly reduced fertility [3]. In terms of educational results, women are also vulnerable; the shift from school to childbirth negatively influences their labour market possibilities. For example, women cover two-thirds and 76 million (over 61%) of the world's 774 million illiterate adults and 123 million illiterate youths respectively and more young people achieve greater educational rates compared to females [4].

According to the IMF, female labour force participation is vital for any country's development as it boosts productivity, increases economic diversification, reduces income inequality and enhances other development outcomes [5]. Studying labour force involvement particularly among females is therefore not only an academic interest, but also has significant implications on gender issues and country's development. The pioneering works of Becker [6] and Mincer [7] on female labour force participation paved way for numerous other scholars to invest on the subject matter. However, despite the identified crucial role provided by females in the provision of labour, their participation rates in the past twenty-five years has declined. Globally this rate has dropped from 51.19% in the year 1990 to a historic low of 47.31% in the year 2019 while in Sub Saharan Africa this rate decreased from 63.99% to 61.29% in the same timeframe as shown on Fig 1 [8].

In developing nations, female's engagement in provision of labour acts as a survival technique to get rid of economic shocks including food insecurity which may affect the household [3]. The lack of or insufficient alternative sources of income from social assistance as well as continuous poverty prevent people from quitting their present jobs [3]. It is for this reason;

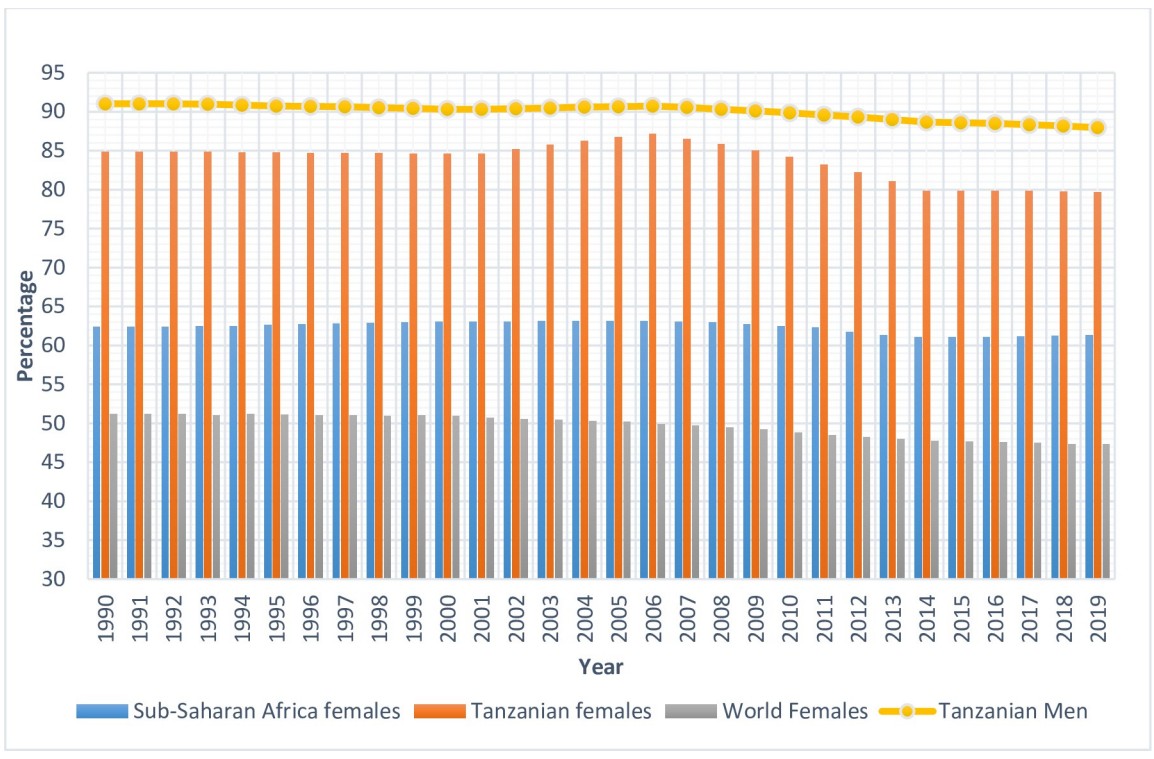

**Fig 1. Labour force participation rates (modelled ILO estimates for ages 15 +).** Source: World Bank [8].

Sub Saharan Africa is home to the first top five nations whose largest share of labour participation rests on females relative to male counterpart. Malawi and Zimbabwe are top in the list constituting over 52% of female share in the provision of labour. Next in the list are Gambia (51%), Liberia about 50.6% followed by Tanzania 50.5% [9].

In Tanzania, female labour force participation rates exhibited higher rates compared to both the global averages and the Sub-Saharan region averages (see Fig 1). Despite the higher rates, participation has experienced successive decline for over a decade; from a record high of 87.16% in 2006 to the lowest rate of 79.69% in 2019 [8]. The decrease is seen as a major blow to the previous improvements in the labour participation rates which showed slightly increases from 84.85% in 1990 to a record high 87.16% in the year 2006 as shown on Fig 1.

In Tanzania, demographic statistics show that female constitutes over half of working age (51.1%) as well as of the total populations (52.04%) respectively [10]. In this sense, females in Tanzania are anticipated to provide more labour relative to males. Nonetheless, over the past two and a half decades, men's participation rate in the provision of labour has been relatively higher compared to that of women (see Fig 1). This is in spite of various government efforts geared towards rising females labour involvement including efforts related to fertility such as paid parental leave with job security, a maximum of two hours a day to breastfeed a child, as well as prenatal and postpartum free medical care for women entitled with maternity benefits [2].

It is worth noting that Tanzania is among the countries with very high fertility rates being 16[th] highest in the world [11]. It is argued that the high fertility rates may reduce female labour force participation as much time is spent on taking care of children [6]. However, others concur that, higher fertility rate increases labour force participation [12]. These contrasting views have not received much attention in the Tanzanian context thus provides impetus to investigate their relationships. Furthermore, from the review empirical studies on fertility and labour supply in Tanzania [13–19], there is a wide gap in the extant literature with focus on this topic and hence it is worth to be conducted for the Tanzanian setting due to its relevance described on the following section.

## 2. Relevance to the literature

Empirical findings regarding female labour force participation are far from conclusive [20–24]. While other studies suggest that participation in the provision of labour by females and fertility are substitutes [25, 26], others suggest that they are complements [21, 27]. On the other hand, other existing evidence suggests that higher education enables the acquisition of skills and expertise necessary for labour force participation [27–29]. Other studies revealed that education decreases the likelihood of entering the labour force [18, 22].

In Africa, a somewhat similar study was conducted by De Jong and Abiba [30] and Solomon and Kimmel [31]. While the former estimated the casual effect of fertility on women's employment in 26 African countries, the latter tested the inverseness of fertility and labour supply for married women in Ethiopia. However, both studies restricted their sample to include only married women. De Jong and Abiba [30], further restricted their sample to use having children aged below or equal to six years as an indicator of fertility. In this regard, it becomes difficult to generalize the findings to women with children above 6 years who may affect the female labour participation decisions. This study bridged the knowledge gap by widening its sample to include women of reproductive age (15–49 years of age) of all marital status and with any number of children. Other similar studies in the region included the ones by Maingi in Kenya [32], Olowa and Adeoti in rural parts of Nigeria [28], Bbaale in Uganda [33], Boateng et al in Ghana [27], Roger and Christian in urban Cameroon [34] as well as Sackey in Ghana [35]. However, the findings of these studies are more likely to be biased due to their

failure (for various reasons including the inability of acquiring stronger instruments) of accounting for the potential endogeneity of fertility in the labour force participation decisions. The current study examined the relationship between fertility and labour force participation with caution due to the possibilities of occurrence of endogeneity issue given the presence of unobservable characteristics like work preference and the decision to have more children [26]. The two can collude as women can choose having both children and work at the same time. Addressing the problem of endogeneity requires the use of valid instrumental variables. Valid instruments are those that correlate with fertility which is an endogenous variable and are not anyhow directly linked to female participation in labour markets. Based on literature review, various instruments of fertility have been used by other researchers. The instruments include multiple first birth, contraceptive legislature, First born sex or the sex of first and the second child and infertility shock reported by a woman [23, 30, 36–38].

At the national level, such studies in the context of Tanzania are lacking. In Tanzania, most studies on fertility and labour force participation focuses on their determinants [14, 15, 17, 18]. While a study by Vavrus [19] focused on the relationship between education and fertility, a study by Bugumba investigated the association of fertility and female empowerment [13]. On the other hand, a study by Lihawa analysed fertility and its effects on health among mothers and children in Tanzania [16]. Addressing this literature gap, this study examined the causal effect of fertility on female labour force participation using the latest public Tanzania Demographic and Health Survey of 2015/2016.

## 3. Methodology

### 3.1 Analytical framework

To incorporate various employments of time and their impact on labour supply choices, Becker [6] broadened the neoclassical labour supply hypothesis. In Becker's framework of time allocation, the family is seen as both an economic division and as a unit that produces various levels of utilities and that an individual in the household can maximize utility by spending time on (1) labour market; (2) production activities within the household; and (3) consumption [39]. In this study, a female's utility is assumed to be maximized through the consumption of goods (C) and leisure (V). This is expressed as;

$$U = u(C, V) \tag{1}$$

However, leisure time is expected to be influenced by the children (F) a female has;

$$V = v(F) \tag{2}$$

Therefore, female's utility relationship shown in equation one can further be estimated as:

$$U = u[C, v(F)] \tag{3}$$

The study assumes that, the total available time (T) for a woman is allocated between labour (Y) and leisure (V). Subject to time constraint, the relationship can be presented as:

$$T = Y + V \tag{4}$$

Apart from the time constraint expressed in Eq (4), females also face expenditures (p) and income (w) constraint. The expenditures are assumed to be spent on consumption of goods whereas income comes from participating in the labour force. Hence, the female's budget

constraint can be expressed as:

$$PC = WY \tag{5}$$

$$PC = W(T - V) \tag{6}$$

$$PC + Wv(F) = WT \tag{7}$$

Subject to budget constraint as shown on Eq (7), the optimal levels of both consumption (C) and leisure (V) can be derived from the optimal number of children (F) as;

$$Max\ U[C, v(F)] \tag{8}$$

The utility maximization problem in (8), can be solved to obtain the optimal consumption (C*) level, leisure (V*) as well as fertility (F*). Eqs 9–11 show the functions above in their reduced forms.

$$C^* = c(p, w, T) \tag{9}$$

$$F^* = f(p, w, T) \tag{10}$$

$$V^*(F^*) = v(p, w, T) \tag{11}$$

From Eq (11), optimal number of children affects optimal leisure time. In this lookout, the optimal number of children must as well influence optimal hours of work or female participation decision. This could be denoted using the following equation:

$$Y^* = (F^*, p, w, T) \tag{12}$$

Since y*, is unobservable representing a latent variable that measures working status, its relationship to labour force participation status can be given as:

$$y = \begin{cases} 1, & \textit{if } y* > 0 \\ 0, & \textit{if } y* \leq 0 \end{cases} \tag{13}$$

From Eq (12), the likelihood of females being in the market for labour (y*) depends on the number of children (*f*), wage attributes (w), price-dimensions (p) time (T). Where $a$, $\delta$, $\beta$, $\rho$, $\kappa$ are parameters that were estimated while $\varepsilon$ is the disturbance term. In this regard, the equation is given as;

$$y^* = a + \delta f + \beta' p + \rho w + \kappa T + \varepsilon \tag{14}$$

Given that this study's objective was to synthesize fertility's influence on labour participation by females, Eq (13) was linked to Eq (14) resulting in the following equation;

$$y = 1[a + \delta f + \beta' p + \rho w + \kappa T + \varepsilon > 0] \tag{15}$$

The probit model can be used to estimate the coefficients including $\delta$. The tendency of a woman to have one extra child may be influenced by factors which determine the female's participation in provision of labour but not observable in the eyes of the researcher. The resultant effect of unobservable covariates is the occurrence of inconsistent and biased findings of $\delta$ emanating from the perfect fertility-disturbance term correlation. Because probit model fails to take into consideration the endogeneity problem, this study employed the instrumental

variable probit (IV-probit) model [32, 40]. The model is specified in Eq 16;

$$f = \alpha'x + \emptyset'z + \upsilon \tag{16}$$

Where, $\phi$ and $\alpha$, are a group of parameters that were estimated, z contains instrumental variables and $\upsilon$ is the disturbance term. Therefore, the IV- probit estimates Eq (16) to obtain the predicted values as shown below:

$$\hat{f} = \hat{\alpha}'x + \hat{\emptyset}'\hat{z} + \hat{\upsilon} \tag{17}$$

Based on Eq (17), $\hat{f}$ is now uncorrelated with $\upsilon$. This implies that it should replace $f$ in Eq (15) to give the model as:

$$y = 1[\delta\hat{f} + \beta'p + \rho w + \kappa T + \varepsilon > 0] \tag{18}$$

Alternatively, one can obtain the residuals from the first stage regression (from Eq 15) and then run the second stage regression with the original endogenous variable, observed confounders and the residuals from the first stage as an added covariate. This relationship may be given as;

$$y = 1[a + \delta f + F* + \beta'p + \rho w + \kappa T + \varepsilon > 0] \tag{19}$$

This approach is known as the two-stage residual inclusion model (2SRI). The addition of a reduced form fertility residual to the estimated labour force participation equation helps to purge the observed relationship between labour force participation and fertility of any effect of the unobservables by allowing fertility to be treated as if it were exogenous during estimation. The inclusion of the fertility residuals (F*) leads to an OLS estimate of the coefficient of fertility that is identical to the one obtained by IV using $\hat{f}$ as an instrument for fertility (see Eq 18). The two-stage residual inclusion (2SRI) corrects for potential sample selection bias that may arise by working with a truncated sample (e.g., female labour force participation age is from 18–60 years but the study ended up with fertile age of 15–49 years).

Assuming the unobserved component is linear in the fertility residual (F*), the addition of an interaction term of the fertility and it's residual (FF*) as a second control variable is sufficient to eliminate endogeneity bias even if the reduced form fertility is heteroscedastic [41, 42]. This approach is known as the Control function approach (CFA). It controls for the effects of neglected non-linear interactions of unobservable variables with determinants of labour force participation and is used to correct both bias due to selection and that emanating from unobserved heterogeneity captured by the interaction of fertility [42]. Based on this the CFA may be specified as;

$$y = 1[a + \delta f + F* + FF* + \beta'p + \rho w + \kappa T + \varepsilon > 0] \tag{20}$$

From Eqs (18), (19) and (20) the study employed contraceptive use, household size and wealth index of households as proxies for price (p). The proxies ensure all households are measured in terms of revenue, access to social facilities and health outcome [21]. In addition, due to the fact that an individual's wage (w) cannot be directly observable, then age and her education status are used as an alternative. This is because, age is used as a measure of experience which is directly proportional to wage and other earnings [21, 43]. In addition, the study employed educational attainment as proxy for wage, due to the fact that, increases in earnings may be caused by higher education attainment and that various education levels differs in terms of payments [43]. Lastly, geographical area, listening to radio, reading newspapers was

used as proxies for time (T). This is due to the fact that their variations might implicitly influence female's time spent on work as well as on leisure.

## 3.2 Instrumenting for fertility

This study used infertility shock, multiple first birth, sex composition of first two children and mother's opinion on ideal number of children as instrumental variables for fertility. These variables were assumed to have no straight impact on the probability of participation of females to provision of labor except indirectly through fertility.

The use of multiple first births has been among the most common instrumental variable ever since their first use by Rosenzweig and Wolpin [44]. It is expected that, multiple first birth increases the aggregate children per woman and thus leading to a rise in the total number she has given birth. Infertility shocks reported by a woman has also been among the most common instrumental variables strategies. It is expected that the infertility shock will highly correlate with fertility due to the fact that the probability of females that are infertile to give birth is minimal.

According to Solomon and Kimmel [31], a valid instrument for fertility should be derived from institutions as well as cultural experiences and norms. Therefore, this study's instrument of the same sex of the first two children of woman (boy-boy or girl-girl) and mother's opinion on ideal family size which is a reflection of cultural experience and norms, and thus is deemed to be credible instruments for fertility. According to James [45] and Sander [46], the use mothers' ideal family size was one among the earliest instruments of fertility in the labour supply. Similarly, according to Angrist and Evans [47], when a woman bears children of the same sex particularly on her first and second birth on the continuum, this increases the likelihood of having the third child as compared to a woman who has given birth to children with different sex in their first and second births. This thus limits analysis of females having more than two children though with the merit of exogeneity of the instrumental variable. In the same continuum, mother's opinion on the suitable size of the family is a proxy for woman's sought aggregate children she might have, and thus it is difficult to argue that they have no effect on labour market behaviour other than via fertility [48].

## 3.3 Source of data

Our study extracted data from the Tanzania Demographic and Health Surveys (DHS) carried-out in 2015–2016. The DHS dataset is a national-wide survey comprising of data on women aged 15–49 concurrently with children they possess. Units surveyed by the DHS 2015–2016, included a total of 13,634 women were intended for interview but at the end of interview, those who finished with all required information totalled at 13,266 which is approximately 97% responsiveness. The definition of variables used in the analysis are described in Table 1.

# 4. Results

## 4.1 Descriptive results

Table 2 presents the socio-economic characteristics of respondents. The results show that of the 13266 respondents used in this study, 24.15% were not working while 75.85% engaged in the labor market through various occupation groups namely professional/technical/managerial, clerical, agricultural-self-employed, agricultural-employee, household and domestic, services, skilled manual and unskilled manual. Similarly, the majority of the respondents (57.59%) had primary education while 26.29% and 1.06% had secondary and higher education respectively. This implies that the literacy rate was about 84.94%. About 46.26% were married

**Table 1. Description of variables.**

| Variable | Description |
|---|---|
| ***Dependent Variable*** | |
| Female Labour Force Participation | 1 if a woman is currently working, 0 otherwise) |
| ***Predictor Variable*** | |
| Fertility | Total number of children ever born to a woman |
| ***Independent Variables*** | |
| Age | Age of a female measured in years |
| Age square | Age of a woman (in years) squared |
| Household size | Number of the person living in the household |
| Education | 1 = no formal education, 0 otherwise |
| | 1 = primary education, 0 otherwise |
| | 1 = secondary education, 0 otherwise |
| | 1 = higher education, 0 otherwise |
| Marital Status | 1 = married, 0 otherwise |
| Wealth Quantiles | Wealth quantiles were measured using the approach that households were given scores based on the number and kinds of consumer goods they own, ranging from a television to a bicycle or car, plus housing characteristics, such as source of drinking water, toilet facilities, and flooring materials. These scores are derived using principal component analysis. National wealth quintiles are compiled by assigning the household score to each usual (de jure) household member, ranking each person in the household population by their score, and then dividing the distribution into five equal categories, each with 20% of the population. Based on this, five categories were formulated such that; |
| | 1 = poorest, 0 otherwise |
| | 1 = poorer, 0 otherwise |
| | 1 = middle, 0 otherwise |
| | 1 = richer, 0 otherwise |
| | 1 = richest, 0 otherwise |
| Place of residence | 1 = lives in rural area, 0 otherwise |
| Contraceptive use | 1 = uses contraceptives, 0 otherwise |
| Husband employment | 1 = employed, 0 otherwise |
| | 1 = husband is self-employed, 0 otherwise |
| ***Instrumental variables*** | |
| Multiple first birth | 1 = first birth is multiple, 0 otherwise |
| Infertility shock | 1 = woman desire not to have more children due to infertility, 0 = menopausal |
| Same gendered children | 1 = first two children are of the same sex, 0 = otherwise |
| Mother's opinion on the ideal family size | A woman's hypothetical ideal number of children to have |

*Source*: Tanzania DHS 2015/16

and 68.75% lived in rural areas while 31.25% were urban residents. This signifies that most of females surveyed were from rural areas as compared to urban areas. With regard to wealth status, the poorest constituted of about 16.16% while the richest quantile constituted about 25.7%.

Similarly, 29.07% used contraceptives as methods of family planning while about 9.93% experienced infertility shocks brought about by infertility problems that faced females during their life processes. These results imply that there is low use of contraceptives as family planning tools in Tanzania while most females do not encounter infertility shocks which may lead to an increase in the rate of population growth if no proper family planning education is

**Table 2. Descriptive statistics.**

| Variable | Measurement | Observations | Percentage (%) |
|---|---|---|---|
| *Categorical variables* | | | |
| Female Labour force Participation | not working | 3,204 | 24.15 |
| | professional/technical/managerial | 418 | 3.15 |
| | clerical | 79 | 0.6 |
| | agricultural—self employed | 5,487 | 41.36 |
| | agricultural—employee | 308 | 2.32 |
| | household and domestic | 630 | 4.75 |
| | services | 453 | 3.41 |
| | skilled manual | 560 | 4.22 |
| | unskilled manual | 2,127 | 16.03 |
| Education | No formal education | 1,998 | 15.06 |
| | Primary education | 7,640 | 57.59 |
| | Secondary education | 3,488 | 26.29 |
| | Higher education | 141 | 1.06 |
| Marital Status | Never in a union | 3,478 | 26.22 |
| | Married | 6,137 | 46.26 |
| | Lives with partner | 2,052 | 15.47 |
| | Not living together | 1,599 | 12.05 |
| Wealth Quantiles | Poorest | 2,144 | 16.16 |
| | Poorer | 2,166 | 16.33 |
| | Middle | 2,438 | 18.38 |
| | Richer | 3,108 | 23.43 |
| | Richest | 3,409 | 25.70 |
| Place of residence | Urban | 4,146 | 31.25 |
| | Rural | 9,120 | 68.75 |
| Contraceptive use | Uses contraceptives | 3,856 | 29.07 |
| | Do not use | 9,410 | 70.93 |
| Infertility shock | Infertile | 1,317 | 9.93 |
| | Not infertile | 11,949 | 90.07 |
| Listens to radio | Listens to radio | 10,321 | 77.80 |
| | Do not listen | 2,945 | 22.20 |
| Reads newspapers | Read newspaper | 5,570 | 41.99 |
| | Do not read | 7,695 | 58.01 |
| Multiple first birth | First birth is multiple | 220 | 2.26 |
| | Not multiple | 9,501 | 97.74 |
| Same gendered children | Same sex children | 3,848 | 29.01 |
| | Not the same | 9,418 | 70.99 |
| *Continuous variables* | *Observations* | *Mean* | *Std. Dev* |
| Fertility | 13,266 | 2.802 | 2.762 |
| Age of a woman | 13,266 | 28.691 | 9.707 |
| Ideal number of children | 13,266 | 4.794 | 2.502 |
| Household size | 13,266 | 6.788 | 3.9 |

*Source*: Authors' computations

provided. This may also affect participation in provision of labour by females since most of the time will be used for rearing children. On the methods used to access information, 77.08% used radio while 41.99% used newspapers. With regard to birth status, 15.08% of the

respondents had one child while 58.19% of the respondents had at least two children. From these, about 2.26% had first multiple births of which 29.01% were of the same sex. On the other hand, 26.72% of the respondents had no children.

The age of females included in the sample averaged at about 28.7 years with a fertility rate of 2.8 which implies that most of the females were youths with an ability of bearing a total of three children. The ideal total children considered by a female was 5 children with an average household size of 6.8 people.

## 4.2 Empirical results

Table 4 displays estimates of structural forms of female labour force participation under different assumptions. Specifically, column (1) shows the standard probit estimates of the structural parameters whereas column (2) gives the IV probit estimates of the structural parameters after taking into account the potential endogeneity of the fertility variable. Column 3 shows the two-stage residual inclusion for potential sample selection bias while column 4 represents the control function approach estimates used to correct both bias due to selection and that emanating from unobserved heterogeneity captured by the interaction of fertility with its residual.

Before presenting the estimations to the outcome of interest, the study sought to test the validity of the instruments used in this study to instrument fertility which was found to be endogenous (Table 4). From the results on Table 3, in the reduced equation, all the instruments were statistically significant at one percent level. This implies that, the instruments correlated with fertility but did not correlate with the outcome variable (participation) and are therefore relevant. In addition, for the case of multiple instruments, the joint F statistics was also greater than 10 [49], indicating that the instruments correlated with fertility. In the participation (FLFP) equation, p -values for infertility shocks, same gendered children and multiple births are insignificant which implies that instruments do not directly explain Labour force participation and are therefore valid. However, the instrumental variable mothers' opinion on ideal number of children was significant at one percent and thus was not part of analysis performed.

From Table 4, probit estimates in column (1) indicated that, with one more child, the likelihood of participation by females in the labour force decreases by about 1.1% holding other factors constant. The observed result from the probit model might be biased due to failure to take into account of the variable–fertility included in the model. In column 2, this study employed the IV probit model with infertility shock, multiple first birth, sex composition of first two children and mother's opinion on ideal number of children as instruments. The IV probit results showed that as an additional child is born fertility decreases labour force participation by a margin of 13% ceteris paribus. Relatively large number of children forces females to increase the time spent on taking care of her children thereby making them reduce the amount of time

**Table 3. Tests of relevance and validity of instruments.**

| Instrumental Variables | Reduced form equation (fertility equation) | Structural equation (FLFP equation) |
|---|---|---|
| Infertility shocks | (P = 0.000) | (P = 0.998) |
| Same gendered children | (P = 0.000) | (P = 0.133) |
| Multiple births | (P = 0.000) | (P = 0.718) |
| Infertility shocks, Same gendered children, Multiple births. | $F(4, 9698) = 124.72$ (P = 0.000) | $\chi^2 (4) = 15.82$ (P = 0.0033) |

*Source*: Authors' computations

**Table 4. Effect of fertility on female labour force participation.**

| Variable name | | Marginal effects | | | |
|---|---|---|---|---|---|
| | | Probit | IV -Probit | 2SRI | CFA |
| | | (1) | (2) | (3) | (4) |
| Fertility | | -0.011*** | -0.130*** | -0.034*** | -0.035*** |
| | | (-0.003) | (-0.041) | (-0.011) | (-0.011) |
| Education | Primary | 0.041*** | 0.092* | 0.026* | 0.025* |
| | | (-0.013) | (-0.05) | (-0.014) | (-0.014) |
| | Secondary | 0.028* | 0.011 | 0.003 | 0.003 |
| | | (-0.016) | (-0.072) | (-0.019) | (-0.019) |
| | Higher | 0.134*** | 0.541** | 0.115*** | 0.113*** |
| | | (-0.025) | (-0.232) | (-0.033) | (-0.034) |
| Husband employment | Employed | 0.019 | -0.005 | 0.001 | -0.002 |
| | | (-0.039) | (-0.15) | (-0.041) | (-0.041) |
| | Self-employed | 0.134*** | 0.410*** | 0.112*** | 0.113*** |
| | | (-0.039) | (-0.15) | (-0.041) | (-0.041) |
| Contraceptive use | | 0.068*** | 0.244*** | 0.066*** | 0.065*** |
| | | (-0.009) | (-0.036) | (-0.009) | (-0.009) |
| Wealth | Richest | -0.019 | -0.127 | -0.038 | -0.036 |
| | | (-0.021) | (-0.085) | (-0.025) | (-0.024) |
| | Richer | -0.041** | -0.136** | -0.041** | -0.038** |
| | | (-0.017) | (-0.062) | -0.018 | -0.018 |
| | Poor | 0.011 | 0.044 | 0.01 | 0.012 |
| | | (-0.015) | (-0.058) | (-0.016) | (-0.015) |
| | Poorer | 0.053*** | 0.214*** | 0.053*** | 0.055*** |
| | | (-0.014) | (-0.0059) | (-0.014) | (-0.014) |
| Married | | -0.071*** | -0.273*** | -0.064*** | -0.070*** |
| | | (-0.01) | (-0.042) | (-0.01) | (-0.01) |
| Rural | | 0.047*** | 0.192*** | 0.053*** | 0.055*** |
| | | (-0.014) | (-0.049) | (-0.014) | (-0.014) |
| Age | | 0.030*** | 0.146*** | 0.038*** | 0.040*** |
| | | (-0.004) | (-0.021) | (-0.006) | (-0.006) |
| Age square | | 0.000*** | -0.001*** | 0.000*** | 0.000*** |
| | | (0.0000) | (0.0000) | (0.0000) | (0.0000) |
| Household size | | 0.004*** | 0.025*** | 0.007*** | 0.007*** |
| | | (-0.001) | (-0.007) | (-0.002) | (-0.002) |
| Fertility residual | | | | 0.026** | 0.031** |
| | | | | (0.011) | (0.012) |
| Fertility interaction | | | | | -0.001 |
| Number of observations | | 9,721 | 9,721 | 9,721 | (0.001) |
| | | | | | 9,721 |
| Wald test of exogeneity (corr = 0): chi2(1) = 5.39 Prob > chi2 = 0.0203 | | | | | |

***, **, * represents 0.01, 0.05 and 0.1 significant levels while figures in parentheses are standard errors.

*Source*: Authors' computations from the Tanzania DHS 2015/16

spent on productive works particularly in the provision of market labour. The result corroborates with previous findings [50, 51]. For instance, a study by Tiwari et al. [50] in India revealed that there was a reduction in participation in the provision of labour force by women

by about 3.5% for women having more than three children compared to those with less children.

The findings further showed that, when we take control for selection bias and endogeneity inherent in the fertility variable, (i.e when two stage residual inclusion function approach is used without the interaction term depicted in the third column) the fertility rate decreases the likelihood of participation by females in provision of labour by a margin of 3.4% as an additional child is born. However, after accounting for the possibility of non-linear interactions of fertility with unobservables as shown in column (4), the findings depicted somewhat similar results to that of the two stage residual inclusion in column (3) whereby the likelihood of participation by females in the provision of labour falls by about 3.5% when an additional child is born, ceteris paribus.

The findings in Table 4 further showed that, besides fertility, there are also other variables that significantly influence participation in the provision of labour by women. These included education, wealth, marital status, age, contraceptive use, household size, and husband's nature of employment and the geographical place of residence. While primary education and higher education were significant in all the used models, secondary education was only significant in the probit model. However, the findings in the four models used in the study showed that as education increases, participation by females increases as well. Education provides skills and knowledge that could be integrated to more opportunities, more productivity that exposes the females further to a wide range of social and economic opportunities and access to resources relative to those with lower level of education [51, 52].

Regarding wealth, the poorer and richer wealth quintiles were found to be significant at 5 percent level in the four models. The results further showed that as wealth increases (except moving from richer to richest wealth group), female labour force participation decreases. Wealthier females tend to participate less in the provision of labour since there is no motive for them to work for paid labour and thus, they choose to allocate more time in caring for the family using the wealth they have. In contrary, poor females tend to participate more in the provision of labour since income that was previously earned motivate them to work more. The results are similar to those of existing findings [51].

In addition, married females had about 6.4–27.2% less likelihood of participation in the provision of labour than those not married as depicted in all the four models specified in the study holding other factors constant. This implies that, being married reduces female participation in the provision of labour since they are considered by employers as child carers and hence may not be available all the time when needed. This is consistent with previous study by Taheri *et al*. [53] in Iran which found similar results. In contrast, the models showed that females whose husbands were self–employed had about 11.2–41% more likelihood of participating in the provision of labour than those who were not self-employed (p<0.01). This can be plausibly explained by the reason that with self-employment of their husbands, females feel part and parcel of family development by providing labour to assist their husbands in generating more income for the family [53].

In all model specifications, variables such as household size, rural place of residence, age and the use of contraceptives were also significant and positive at one percent level. With larger household sizes, females are obliged to provide more labour to generate more income to feed the ever-increasing needs of the household members and thus allocate more time in working for the household rather than house-works [51].

The coefficient for residence (rural versus urban) showed a positive relationship with labour participation. Females residing in rural areas had higher probability ranging from 4.7–19.2% of participating in provision of market labour than their urban counterparts. This is explained by the prevalence of extended families in rural areas relative to urban areas which necessitates

women to work more for paid works to earn income that could help in taking care of her children as well as relatives living with her. The result corroborates those by Taheri *et al.*, [53], Nsanja, [52] and Majbour, [54].

Female's age showed an inverted U-shaped relationship with labour participation. Initially, as a female's age increase by one year, the probability of participation in the provision of labour increases by about 14.6%. Later on, beyond the age of 73 years–the turning point, the probability of participation starts to decrease by about 0.1% for each successive year. The reason for this is that relatively younger females tend to have more access to opportunities given less time spent on caring children since they tend to have few children than the older ones who tend to have more children and thus spend much time in taking care of families at home. Similarly, older females are responsible for other more family duties including taking care of elders which make them allocate more time for house chores, no time to search for opportunities and education unlike younger females. The findings support those of Nsanja [52] in Malawi and Wang *et al.* [51] in China.

The coefficient on the fertility residual in column (3) was significant and positive (p<0.05) which implied that correction for sample selection bias was necessary. In contrast, the interaction term in the control function approach (column 4) was found to be negative and statistically insignificant indicating the absence of unobserved heterogeneity on fertility and participation decisions. The Wald test of exogeneity was also significant at five percent level indicating that fertility was truly endogenous and that the use of instrumental variable probit (column 2) was therefore necessary.

## 5. Conclusion

This paper attempted to empirically establish the nexus between fertility and participation in the provision of labour by females through utilizing the 2015/16 Tanzania demographic and health survey data. Instrumental variable (IV) probit was employed in the analysis while infertility shock, multiple first birth, composition of sex on the first two children and mother's opinion on ideal number of children were used as instrumental variables. The study further used a two stage residual inclusion and an interaction analysis to further omit the inconsistent and biased results that could be obtained emanating from endogeneity and selection bias of the sample used in the study.

Findings in this study further indicated that failure to account for the problem of endogeneity of fertility variable and bias due to selection of the sample in the participation equation results in underestimation of the results as depicted on the results for the probit model under column 1. By taking care of these problems the results of fertility variable increased to about 3 times relative to that of the standard probit results. The negative influence of fertility on participation observed in all models employed in this study is consistent to the findings by Lim [39], Prasetyoputra, Irianti, and Sasimartoyo [40], as well as Sackey [35] in Malaysia, Indonesia and Ghana respectively.

However, this is was contrary to the findings by Kabubo-Mairara *et al.* [26] which found both positive and negative influence of fertility by employing probit and IV–probit models respectively. The study findings also contradict the findings of Rondinelli and Zizza [44] which found that, through the use of probit model, fertility negatively influences labour force participation by females; however, after accounting for endogeneity the negative relationship disappeared.

Methodologically, the study did not take into account the possibility of endogeneity of some variables including education in the participation equation resulting from the likely reverse causality between variables of interest that were entered in the estimated model as well

as those that could not be captured by the model including variables like entrepreneurial capabilities that may be present to some people that could concurrently affect both fertility and participation. In this lookout, the study acknowledges the need for deeper analysis on this topic by including more variables, employing other models and a wider coverage particularly cross-country analysis to take into account heterogeneities in the multi-country settings thereby resulting into a more robust and replicable results.

To guide public policy, the study recommends more emphasis on education attainment since this has been verified as a tool that ignites further female participation in the provision of labour in economic enhancing activities. This could further help the country boosting its economy through employing more human capital with required skills based on gender balance with more emphasis on females. This can be done by strengthening and extending more funds allocated to financing education by considering females students who are more vulnerable to socio-economic shocks including poverty and access to resources in Tanzania due to cultural aspects that tend to favour males. In addition, the study recommends for the provision of vocational training that will impart life's soft skills to females who dropped out of school owing to various reasons including unfriendly environment and pregnancies. Lastly, this paper points the need for policy makers to direct their policies at improving family planning services particularly the use of contraceptives since they have shown to have positive impact on participation by females in the provision of labour in the country that will further enhance their contribution in the development of the country.

## Supporting information

**S1 Appendix. Econometric results.**
(DOCX)

## Author Contributions

**Conceptualization:** Aidan Laurent Sunday.

**Data curation:** Aidan Laurent Sunday.

**Formal analysis:** Aidan Laurent Sunday, Robert Michael Lihawa.

**Investigation:** Aidan Laurent Sunday, Eliaza Mkuna.

**Methodology:** Aidan Laurent Sunday.

**Resources:** Robert Michael Lihawa, Eliaza Mkuna.

**Supervision:** Robert Michael Lihawa, Eliaza Mkuna.

**Writing – original draft:** Aidan Laurent Sunday, Robert Michael Lihawa.

**Writing – review & editing:** Robert Michael Lihawa, Eliaza Mkuna.

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
