## [Decision Letter · Decision Letter 0]

23 Jan 2023

PONE-D-22-31164THE EFFECT OF FERTILITY ON FEMALE LABOUR FORCE PARTICIPATION IN TANZANIAPLOS ONE

Dear Dr. Mkuna,

Thank you for submitting your manuscript to PLOS ONE. After careful consideration, we feel that it has merit but does not fully meet PLOS ONE’s publication criteria as it currently stands. Therefore, we invite you to submit a revised version of the manuscript that addresses the points raised during the review process.

We look forward to receiving your revised manuscript.

Kind regards,

José Alberto Molina

Academic Editor

PLOS ONE

Journal Requirements:

Reviewers' comments:

Reviewer's Responses to Questions

**Comments to the Author**

1. Is the manuscript technically sound, and do the data support the conclusions?

Reviewer #1: Yes

Reviewer #2: Partly

2. Has the statistical analysis been performed appropriately and rigorously? 

Reviewer #1: No

Reviewer #2: No

3. Have the authors made all data underlying the findings in their manuscript fully available?

Reviewer #1: No

Reviewer #2: Yes

4. Is the manuscript presented in an intelligible fashion and written in standard English?

Reviewer #1: Yes

Reviewer #2: Yes

5. Review Comments to the Author

Reviewer #1: This study examines a very interesting and valuable topic and applies an IV method to tackle the endogeneity issue of fertility decisions.

However, I found the discussion of IV validity in general missing. Among the instruments, the subjective instrument -- Mother’s opinion on the ideal family size -- can especially be argued to be endogeneous as women's personality or preference directly impacts both their fertility and labor supply decisions, which violates the validity assumption. Performing the overidentication test would help provide inferences for the validity of instruments. Reporting weak identification test results is also standard while using multiple IVs, although first-stage F statistics provide references for instrument relevance. Overall, IVs deserve more careful analyses.

Regarding the analytical framework, I found the notation inconsistent and confusing sometimes. Among others, C sometimes denotes time allocated to work, sometimes means consumption. Moreover, the analytical framework does not add much to the discussion nor guide the empirical analysis. Latent variable is discussed but not applied in empirical model -- it may make a difference if female labor supply at the intensive margin is studied as an outcome variable.

As briefly discussed by the authors in the paper, some chosen IVs only work for women with at least two children. Would including and excluding these IVs affect the sample size and your results differently? What is the proportion of women with at least two children? Some descriptive statistics would help.

Reviewer #2: This paper studies the link between female labour force participation (LFP) and fertility in Tanzania in 2015-2016 period. The empirical strategy exploits cross-sectional variation in fertility and finds that additional children reduce female labour force participation. This is consistent with prior literature documenting similar relationship.

My comments below mostly focus on framing the paper's contribution, and are presented in the order they came up as I read:

1. I wonder what is particular about the Tanzanian context that would motivate investigating the relationship only there, compared to other African contexts. Even if female LFP is the highest there as depicted in Figure 1, this doesn't mean the relationship between fertility and LFP is quite particular to Tanzania. I think it would be more interesting and much more enriching to see a replication of same exercise in other African countries.

2. I think the "related literature" section could be more clear about what the exact contribution of the paper is. So, it would be helpful for framing the paper's contribution more clearly (is the contribution of this paper causal identification of the relationship between fertility and LFP, and/or is the contribution the setting - in Tanzania?). Have there been other papers that have examined the same relationship in Africa? What were their results?

3. It would be helpful to provide some graphical evidence before looking at regressions. For example, a graph of say, mean children ever born and average female LFP in given province/district.

4. Data: is there only 1 DHS survey for Tanzania? This makes a big limitation for the study, by not being able to track the relationship over time. Is it possible to supplement these results with other survey data from Afrobarometer and other World Bank household surveys?

5. As a result, the empirical specification would follow a cross section design, with inability to control for province or district fixed effects, since there is no time variation. The use of IV is quite important to control for reverse feedback and other unobservables, but then the author mentioned it used "two stage residual inclusion function approach" without justifying its importance and how it improves upon profit IV. All methodologies used to estimate the relationship should be carefully elaborated in the empirical specification section. There is also no discussion on the exclusion restriction of IV.

6. Results in Table 4 are quite puzzling especially with regards to magnitudes. In column 2, the estimated coefficient of the IV is quite very large compared to OLS, I wonder why? No explanation was given. I also don't understand the point of using two stage residual inclusion in column 3 and non-linear interactions

of fertility with unobservables as shown in column 4. How this improves upon profit IV, which is supposed to correct for non-observables by default.

7. Also, based on table 2 of descriptive statistics, I see we have an unbalanced dataset. I wonder if the addition of controls led to a drop of sample size and affected the estimated coefficients. What would happen if we estimated the equation without controls?

8. How about non-linear relationship? I would like to see interactions of the fertility variable with individual characteristics to explore their conditional effects.

Minor comments:

1- Results table must include the number of observations for each model and adjusted R-squared to evaluate the predicative power of the model.

2- A short paragraph should be included in the introduction that defines the sections of the paper.

6. PLOS authors have the option to publish the peer review history of their article (what does this mean?). If published, this will include your full peer review and any attached files.

Reviewer #1: No

Reviewer #2: No

---

## [Author Response · Author response to Decision Letter 0]

26 Apr 2023

Reviewers' comments:

Reviewer's Responses to Questions

Comments to the Author

1. Is the manuscript technically sound, and do the data support the conclusions?

Reviewer #1: Yes

Reviewer #2: Partly

2. Has the statistical analysis been performed appropriately and rigorously?

Reviewer #1: No

Reviewer #2: No

3. Have the authors made all data underlying the findings in their manuscript fully available?

Reviewer #1: No

Reviewer #2: Yes

4. Is the manuscript presented in an intelligible fashion and written in standard English?

Reviewer #1: Yes

Reviewer #2: Yes

5. Review Comments to the Author

Reviewer #1: This study examines a very interesting and valuable topic and applies an IV method to tackle the endogeneity issue of fertility decisions.

1. However, I found the discussion of IV validity in general missing. Among the instruments, the subjective instrument -- Mother’s opinion on the ideal family size -- can especially be argued to be endogeneous as women's personality or preference directly impacts both their fertility and labor supply decisions, which violates the validity assumption. Performing the overidentication test would help provide inferences for the validity of instruments. Reporting weak identification test results is also standard while using multiple IVs, although first-stage F statistics provide references for instrument relevance. Overall, IVs deserve more careful analyses.

RESPONSES:

The discussion of IV validity has been added and improved as per suggestion (see section 4.2 and table 3). Also, the justification for the use of Mother’s opinion on the ideal family size has been justified (the last paragraph of section 3.2, line xxx). In addition, the test of relevance and validity of the instruments was done (table 3) and all the instruments were valid and relevant, hence Mother’s opinion on the ideal family size had no influence on labour supply decisions except through fertility. 

2. Regarding the analytical framework, I found the notation inconsistent and confusing sometimes. Among others, C sometimes denotes time allocated to work, sometimes means consumption. 

RESPONSES:

The letters have been changed to avoid confusion. Letter C now denotes Consumption whereas time is denoted by letter T.

3.Moreover, the analytical framework does not add much to the discussion nor guide the empirical analysis. Latent variable is discussed but not applied in empirical model -- it may make a difference if female labor supply at the intensive margin is studied as an outcome variable.

RESPONSES:

Analytical framework has been improved and more description on the methods used has been added (See section 3.1). The latent variable explained in the study represents an unobservable that implies the likelihood of females being in the labour market. Therefore, the dependent variable in this case labour force participation has been studied as an outcome variable as described from equation 12-18.

4. As briefly discussed by the authors in the paper, some chosen IVs only work for women with at least two children. Would including and excluding these IVs affect the sample size and your results differently? What is the proportion of women with at least two children? Some descriptive statistics would help.

RESPONSES:

Excluding the instrumental variables affects the sample sizes and results differently. By including the instrumental variables (multiple first birth and the sex composition of first two children) the number of observations reduces from 13,266 to 9,720. However, excluding them will lead to biased estimates as the endogeneity of fertility in the labour force participation equation will not be accounted. Additionally, the descriptive statistics of women with at least two children has been added. 

Reviewer #2: This paper studies the link between female labour force participation (LFP) and fertility in Tanzania in 2015-2016 period. The empirical strategy exploits cross-sectional variation in fertility and finds that additional children reduce female labour force participation. This is consistent with prior literature documenting similar relationship.

My comments below mostly focus on framing the paper's contribution, and are presented in the order they came up as I read:

1. I wonder what is particular about the Tanzanian context that would motivate investigating the relationship only there, compared to other African contexts. Even if female LFP is the highest there as depicted in Figure 1, this doesn't mean the relationship between fertility and LFP is quite particular to Tanzania. I think it would be more interesting and much more enriching to see a replication of same exercise in other African countries.

RESPONSES:

The justification for the inclusion of Tanzania as the case study has been added with relevant literature. As stated in the last paragraph of section 2, while there are a number of studies regarding fertility and labour force participation in many African countries, in Tanzania such studies are lacking. This is despite the fact that Tanzania is among the countries with very high fertility and labour force participation rates in the world (see figure 1). 

2. I think the "related literature" section could be more clear about what the exact contribution of the paper is. So, it would be helpful for framing the paper's contribution more clearly (is the contribution of this paper causal identification of the relationship between fertility and LFP, and/or is the contribution the setting - in Tanzania?). Have there been other papers that have examined the same relationship in Africa? What were their results?

RESPONSES:

The contribution of the paper has been improved in the relevant literature section. In addition, more discussion on the contribution of the study in relation to other studies in the African region has been added. As stated in the last paragraph of section 2, the study is to the best of the researchers’ knowledge; the first to examine the causal effect of fertility and LFP in Tanzania. 

3. It would be helpful to provide some graphical evidence before looking at regressions. For example, a graph of say, mean children ever born and average female LFP in given province/district.

RESPONSES:

The summary of the information suggested have been presented in table 2 

4. Data: is there only 1 DHS survey for Tanzania? This makes a big limitation for the study, by not being able to track the relationship over time. Is it possible to supplement these results with other survey data from Afrobarometer and other World Bank household surveys?

RESPONSES:

The DHS survey used herewith is the latest public available dataset in Tanzania. Afrobarometer and World bank data had some inconsistency in terms of specific variables of interest. However, we are planning to write a follow up paper by integrating the upcoming Tanzania DHS that might be released by the government in future and we might use different approaches (in terms of the methodologies). 

5. As a result, the empirical specification would follow a cross section design, with inability to control for province or district fixed effects, since there is no time variation. The use of IV is quite important to control for reverse feedback and other unobservables, but then the author mentioned it used "two stage residual inclusion function approach" without justifying its importance and how it improves upon profit IV. All methodologies used to estimate the relationship should be carefully elaborated in the empirical specification section. There is also no discussion on the exclusion restriction of IV.

RESPONSES:

The empirical specification section has been improved. Specifically, the use of the "two stage residual inclusion and the control function approach" has also been justified 

6. Results in Table 4 are quite puzzling especially with regards to magnitudes. In column 2, the estimated coefficient of the IV is quite very large compared to OLS, I wonder why? No explanation was given. I also don't understand the point of using two stage residual inclusion in column 3 and non-linear interactions of fertility with unobservables as shown in column 4. How this improves upon profit IV, which is supposed to correct for non-observables by default.

RESPONSES:

7. Also, based on table 2 of descriptive statistics, I see we have an unbalanced dataset. I wonder if the addition of controls led to a drop of sample size and affected the estimated coefficients. What would happen if we estimated the equation without controls?

RESPONSES:

We tested for the same and found insignificant results without control 

8. How about non-linear relationship? I would like to see interactions of the fertility variable with individual characteristics to explore their conditional effects.

RESPONSES:

We tested for non-linearity relationship and the results were not significant.

Minor comments:

1- Results table must include the number of observations for each model and adjusted R-squared to evaluate the predicative power of the model.

RESPONSES

The total number of observations has been indicated in Table 2 i.e. 13,266.

2- A short paragraph should be included in the introduction that defines the sections of the paper.

RESPONSES

The comments have been addressed based on reviewer’s suggestions. A paragraph on the introduction of the paper’s sections has been added. ‘The rest of the paper is structured with the following sections; relevance to the literature, methodology, results and conclusions’.

---

## [Decision Letter · Decision Letter 1]

26 Jun 2023

PONE-D-22-31164R1THE EFFECT OF FERTILITY ON FEMALE LABOUR FORCE PARTICIPATION IN TANZANIAPLOS ONE

Dear Dr. Mkuna,

Thank you for submitting your manuscript to PLOS ONE. After careful consideration, we feel that it has merit but does not fully meet PLOS ONE’s publication criteria as it currently stands. Therefore, we invite you to submit a revised version of the manuscript that addresses the points raised during the review process.

We look forward to receiving your revised manuscript.

Kind regards,

José Alberto Molina

Academic Editor

PLOS ONE

Reviewers' comments:

Reviewer's Responses to Questions

**Comments to the Author**

1. If the authors have adequately addressed your comments raised in a previous round of review and you feel that this manuscript is now acceptable for publication, you may indicate that here to bypass the “Comments to the Author” section, enter your conflict of interest statement in the “Confidential to Editor” section, and submit your "Accept" recommendation.

Reviewer #2: All comments have been addressed

Reviewer #3: All comments have been addressed

2. Is the manuscript technically sound, and do the data support the conclusions?

Reviewer #2: Yes

Reviewer #3: Partly

3. Has the statistical analysis been performed appropriately and rigorously? 

Reviewer #2: Yes

Reviewer #3: No

4. Have the authors made all data underlying the findings in their manuscript fully available?

Reviewer #2: Yes

Reviewer #3: Yes

5. Is the manuscript presented in an intelligible fashion and written in standard English?

Reviewer #2: Yes

Reviewer #3: Yes

6. Review Comments to the Author

Reviewer #2: The authors have addressed all the comments raised in their revised manuscript. I have no further comments.

Reviewer #3: This is an interesting paper dealing with a somewhat overstudied and quite obvious topic. However, as the authors claim, it is a topic that has not been fully studied in Tanzania and has not been approached 'causally' before in Sub-Saharan Africa. The latter can be the main contribution of this paper.

In this revised version, the authors have done important work in addressing some of the comments from the previous reviewers. However, I believe more work is needed, especially in contextualizing the data and its characteristics, in order to enhance the persuasiveness of the results and convince the audience.

I will provide some comments here on aspects I believe should be reworked in the paper to achieve a satisfactory status for publication.

MAJOR COMMENTS:

In Figure 1, the authors provide a chart comparing the percentage of labor force participation (LFP) for Tanzanian women compared to Tanzanian men and women in the rest of Sub-Saharan African countries and the world as a whole. For the sake of consistency, this chart needs a note explaining the age range included and the type of LFP being taken into account (e.g., formal only, formal + an estimation of informal). However, perhaps more importantly, especially for an international audience not familiar with Tanzania, the authors need to provide some context on why there has been a 5 percentage point drop in LFP since 2010. Some contextualization would be helpful, especially considering that the data studied is from 2015-16.

Another important aspect to consider, which may require further clarification, is the characteristic of labor participation. In Section 3.3, the authors present the description of the variables in a table with few details. However, being considered as 'working' can have various meanings. Is there any way of knowing the distribution of labor for women? For example, what fraction were engaged in part-time labor or gainfully employed? Were those in rural areas primarily working at the family and subsistence level or employed elsewhere? I am not sure if the authors have access to all this information, but I believe the Demographic and Health Survey (DHS) provides an explanation of the variables, and including some details here would be beneficial for readers to understand the nature of the labor being discussed.

Regarding Table 2 (descriptives), some values are difficult to interpret, especially for readers unfamiliar with Tanzania. What do the Wealth Quantiles measure? Why are these quintiles so unbalanced? Moreover, why are 25.7% of the women interviewed considered to be among the richest quantile? I assume this may sound somewhat unusual to other readers, and providing more details would help in understanding the representativeness of the sample.

Similarly, in the fertility categories, it is unclear why there are only three categories, including one with 2 or more children, in a country where we know fertility rates are high. What is the rationale behind this categorization? Why not treat it as a continuous variable? Alternatively, why not have a category for a number of children above the mean? Another option could be to establish a category for 3 children and more, aligning with the international literature that often distinguishes between the desirable norm of 2 children and 3 or more children in many countries.

Lastly, I am surprised to see that, among all the answers the authors provided to the previous reviewers, they did not address what may be the most important comment. Allow me to paste the previous reviewer's comment regarding Table 4 here:

“Results in Table 4 are quite puzzling especially with regards to magnitudes. In column 2, the estimated coefficient of the IV is quite very large compared to OLS, I wonder why? No explanation was given. I also don't understand the point of using two stage residual inclusion in column 3 and non-linear interactions of fertility with unobservables as shown in column 4. How this improves upon profit IV, which is supposed to correct for non-observables by default”

Why did the authors provide no answer or make modifications to address this methodological and empirical claim? This is an important issue that needs clarification, especially considering the potential contribution of the paper's causal approach.

MINOR COMMENTS: The visual and formal aspects of the chart (Figure 1) and the regression tables need improvement. Most importantly, as mentioned by a previous reviewer, the regression tables should always report the number of cases (N) in each regression model as per convention. It is not sufficient to state that this information has already been reported in the descriptive table. Readers should have access to this information directly in the table

7. PLOS authors have the option to publish the peer review history of their article (what does this mean?). If published, this will include your full peer review and any attached files.

Reviewer #2: No

Reviewer #3: **Yes: **Gabriel Brea-Martínez

---

## [Author Response · Author response to Decision Letter 1]

10 Aug 2023

MAJOR COMMENTS:

1. In Figure 1, the authors provide a chart comparing the percentage of labor force participation (LFP) for Tanzanian women compared to Tanzanian men and women in the rest of Sub-Saharan African countries and the world as a whole. For the sake of consistency, this chart needs a note explaining the age range included and the type of LFP being taken into account (e.g., formal only, formal + an estimation of informal). 

Responses

The comment has been considered by explaining the age range recommended by the reviewer. See Figure 1 which presents the Labor force participation rates (modeled ILO estimate for ages 15+). 

2. Another important aspect to consider, which may require further clarification, is the characteristic of labor participation. In Section 3.3, the authors present the description of the variables in a table with few details. However, being considered as 'working' can have various meanings. Is there any way of knowing the distribution of labor for women? For example, what fraction were engaged in part-time labor or gainfully employed? Were those in rural areas primarily working at the family and subsistence level or employed elsewhere? I am not sure if the authors have access to all this information, but I believe the Demographic and Health Survey (DHS) provides an explanation of the variables, and including some details here would be beneficial for readers to understand the nature of the labor being discussed.

Responses

Table 2 has been improved by adding more categories by the distribution of labor for women as recommended and more improvement on the descriptive statistics. 

3. Regarding Table 2 (descriptives), some values are difficult to interpret, especially for readers unfamiliar with Tanzania. What do the Wealth Quantiles measure? Why are these quintiles so unbalanced? Moreover, why are 25.7% of the women interviewed considered to be among the richest quantile? I assume this may sound somewhat unusual to other readers, and providing more details would help in understanding the representativeness of the sample.

Responses

Based on the DHS dataset, the wealth index was measured using the approach that households were given scores based on the number and kinds of consumer goods they own, ranging from a television to a bicycle or car, plus housing characteristics, such as source of drinking water, toilet facilities, and flooring materials. These scores are derived using principal component analysis. National wealth quintiles are compiled by assigning the household score to each usual (de jure) household member, ranking each person in the household population by their score, and then dividing the distribution into five equal categories, each with 20% of the population. In addition, this information was added to the definitions of variables section.

4. Similarly, in the fertility categories, it is unclear why there are only three categories, including one with 2 or more children, in a country where we know fertility rates are high. What is the rationale behind this categorization? Why not treat it as a continuous variable? Alternatively, why not have a category for the number of children above the mean? Another option could be to establish a category for 3 children and more, aligning with the international literature that often distinguishes between the desirable norm of 2 children and 3 or more children in many countries.

Responses

Fertility was treated as a continuous variable (See page 12). 

5. Lastly, I am surprised to see that, among all the answers the authors provided to the previous reviewers, they did not address what may be the most important comment. Allow me to paste the previous reviewer's comment regarding Table 4 here:“Results in Table 4 are quite puzzling especially with regards to magnitudes. In column 2, the estimated coefficient of the IV is quite very large compared to OLS, I wonder ‘[why? No explanation was given. I also don't understand the point of using two stage residual inclusion in column 3 and non-linear interactions of fertility with unobservables as shown in column 4. How this improves upon profit IV, which is supposed to correct for non-observables by default”Why did the authors provide no answer or make modifications to address this methodological and empirical claim? This is an important issue that needs clarification, especially considering the potential contribution of the paper's causal approach.

Responses

The study displayed the standard probit model results in equation 1 (that failed to take into consideration several methodological problems). Based on that, the instrumental variable probit (IV-probit) model was used to correct the endogeneity problem in column 2. In addition, the two-stage residual inclusion model (2SRI) in column 3 helped to purge the observed relationship between labour force participation and fertility of any effect of the unobservables by allowing fertility to be treated as if it were exogenous during estimation. The inclusion of the fertility residuals (F*) leads to an OLS estimate of the coefficient of fertility that is identical to the one obtained by IV using ˆf as an instrument for fertility. The two-stage residual inclusion (2SRI) corrects for potential sample selection bias that may arise by working with a truncated sample (e.g., female labour force participation age is from 18-60 years but the study ended up with fertile age of 15-49 years). On the other hand, the control function approach (CFA) in column 4 controls for the effects of neglected non-linear interactions of unobservable variables with determinants of labour force participation and is used to correct both bias due to selection and that emanating from unobserved heterogeneity captured by the interaction of fertility. 

In this regard, the magnitudes of the coefficients of the IV probit in column 2 may not necessarily be closer (or the same) to the 2SRI and the CFA as the latter approaches improved upon the IV probit.

6. MINOR COMMENTS: The visual and formal aspects of the chart (Figure 1) and the regression tables need improvement. Most importantly, as mentioned by a previous reviewer, the regression tables should always report the number of cases (N) in each regression model as per convention. It is not sufficient to state that this information has already been reported in the descriptive table. Readers should have access to this information directly in the table

Responses

The number of observations has been added at the end of the table for each regression model.

---

## [Decision Letter · Decision Letter 2]

13 Sep 2023

THE EFFECT OF FERTILITY ON FEMALE LABOUR FORCE PARTICIPATION IN TANZANIA

PONE-D-22-31164R2

Dear Dr. Mkuna,

We’re pleased to inform you that your manuscript has been judged scientifically suitable for publication and will be formally accepted for publication once it meets all outstanding technical requirements.

Kind regards,

José Alberto Molina

Academic Editor

PLOS ONE

Additional Editor Comments (optional):

Reviewers' comments:

Reviewer's Responses to Questions

**Comments to the Author**

1. If the authors have adequately addressed your comments raised in a previous round of review and you feel that this manuscript is now acceptable for publication, you may indicate that here to bypass the “Comments to the Author” section, enter your conflict of interest statement in the “Confidential to Editor” section, and submit your "Accept" recommendation.

Reviewer #3: All comments have been addressed

2. Is the manuscript technically sound, and do the data support the conclusions?

Reviewer #3: Partly

3. Has the statistical analysis been performed appropriately and rigorously? 

Reviewer #3: Yes

4. Have the authors made all data underlying the findings in their manuscript fully available?

Reviewer #3: Yes

5. Is the manuscript presented in an intelligible fashion and written in standard English?

Reviewer #3: Yes

6. Review Comments to the Author

Reviewer #3: The authors have answered all my previous questions, and also nuanced most oft their claims. In my opinion the paper is now ready to be published.

7. PLOS authors have the option to publish the peer review history of their article (what does this mean?). If published, this will include your full peer review and any attached files.

Reviewer #3: **Yes: **Gabriel Brea-Martinez

---

## [Editor Report · Acceptance letter]

19 Dec 2023

PONE-D-22-31164R2 

PLOS ONE

Dear Dr. Mkuna, 

I'm pleased to inform you that your manuscript has been deemed suitable for publication in PLOS ONE. Congratulations! Your manuscript is now being handed over to our production team.

Kind regards, 

on behalf of

Professor José Alberto Molina 

Academic Editor

PLOS ONE